# Current Treatment Strategies for Non-Small-Cell Lung Cancer with Comorbid Interstitial Pneumonia

**DOI:** 10.3390/cancers13163979

**Published:** 2021-08-06

**Authors:** Satoshi Ikeda, Terufumi Kato, Hirotsugu Kenmotsu, Akimasa Sekine, Tomohisa Baba, Takashi Ogura

**Affiliations:** 1Kanagawa Cardiovascular and Respiratory Center, Department of Respiratory Medicine, 6-16-1, Tomioka-higashi, Kanazawa-ku, Kanagawa 236-0051, Japan; sekine@kanagawa-junko.jp (A.S.); baba@kanagawa-junko.jp (T.B.); ogura@kanagawa-junko.jp (T.O.); 2Kanagawa Cancer Center, Department of Thoracic Oncology, 2-3-2, Nakao, Asahi-ku, Kanagawa 241-8515, Japan; katote@kcch.jp; 3Division of Thoracic Oncology, Shizuoka Cancer Center, 1007, Shimonagakubo, Nagaizumi-cho, Shizuoka 411-8777, Japan; h.kenmotsu@scchr.jp

**Keywords:** non-small-cell lung cancer, interstitial pneumonia, idiopathic pulmonary fibrosis, acute exacerbation, pneumonitis, cytotoxic drug, immune checkpoint inhibitor, surgery, perioperative treatment, radiation therapy

## Abstract

**Simple Summary:**

Interstitial pneumonia is a poor prognostic comorbidity in patients with non-small-cell lung cancer. No matter how effective the treatment for lung cancer is, once an acute exacerbation of pre-existing interstitial pneumonia occurs, it will be fatal to the patient at a high rate. Therefore, it is important to choose a therapy that is unlikely to induce acute exacerbation of interstitial pneumonia. In this review article, we summarize the current evidence on pharmacotherapy, surgery and perioperative treatment, and radiation therapy for non-small-cell lung cancer with comorbid interstitial pneumonia, and discuss future perspectives.

**Abstract:**

Of patients with advanced non-small-cell lung cancer (NSCLC), 5–10% have interstitial pneumonia (IP) at the time of diagnosis. To avoid fatal acute exacerbations of pre-existing IP, appropriate patient selection and low-risk treatment choices are warranted. Risk factors for acute exacerbation of pre-existing IP with cytotoxic drugs include honeycomb lungs on computed tomography (CT) and low forced vital capacity, but risk factors with immune checkpoint inhibitors (ICIs) have not been fully investigated. For advanced or recurrent NSCLC with comorbid IP, carboplatin plus nanoparticle albumin-bound paclitaxel is the standard of care for first-line treatment, but second-line or later treatment has not been established. ICI holds great promise for long-term survival, but many challenges remain, including safety and appropriate patient selection. Since the indications for pharmacotherapy and radiotherapy for NSCLC with comorbid IP are quite limited, surgical resection should be considered as much as possible for patients with operable stages. A scoring system has been reported to predict the risk of postoperative acute exacerbation of pre-existing IP, but perioperative treatment has not been established. In the future, it is necessary to accumulate more cases and conduct further research, not only in Japan but also worldwide.

## 1. Introduction

Of patients with advanced non-small-cell lung cancer (NSCLC), 5–10% have interstitial pneumonia (IP) at the time of diagnosis and have a worse prognosis than those without IP [1]. Meanwhile, the complication rate of lung cancer in patients with idiopathic IPs (IIPs) is also 7–14 times higher than that in patients without IIPs. Among IIPs, the frequency of lung cancer complications varies, with Kreuter et al. reporting 15.8% for idiopathic pulmonary fibrosis (IPF), 6.3% for nonspecific interstitial pneumonia (NSIP), and 5.6% for cryptogenic organizing pneumonia [2]. In a large-scale epidemiological study of IPF patients in Japan (Hokkaido Study), 11% of all deaths were due to lung cancer [3].

The common risk factors reported for the development of IIPs and lung cancer include smoking, environmental and occupational exposure to toxic substances, bacterial and viral infections, and chronic tissue damage [4,5]. In addition, common genetic alterations in the pathogenesis of lung cancer and IP (especially IPF) have been reported, including microsatellite instability, loss of heterozygosity, p53 mutations, and fragile histidine triad mutations [6,7]. Whole Exome Sequencing analysis of 296 Japanese patients with lung adenocarcinomas, including 54 patients with IP, indicated that the pulmonary surfactant gene mutations are specific for adenocarcinoma with IP, and that those with these mutations have poor prognosis [8]. Furthermore, shortening of telomere length and expression of telomerase, which are associated with cancer development, progression, and proliferation, have also been observed in familial and solitary IPF [9].

In the treatment of NSCLC with comorbid IP, it is necessary to consider not only the prognosis of NSCLC but also the prognosis of comorbid IP itself. In the Hokkaido Study, the median overall survival (OS) of IPF patients was 35 months, and the most common cause of death was acute exacerbation (40%) [3]. Patients with IPF develop acute exacerbation at a frequency of 10–15% per year during their natural course [10,11]. In addition, patients with idiopathic NSIP or IP associated with collagen vascular disease, other than IPF, have also been reported to develop acute exacerbations, with a frequency of 3–5% per year [12]. Acute exacerbation is a fatal condition, and the mortality rate of the first acute exacerbation is as high as about 30–50%. Even if improvement is achieved with treatment with corticosteroids, many patients have residual impaired pulmonary function and hypoxemia, and the subsequent prognosis is poor.

According to the International Working Group Report on acute exacerbation of IPF in 2016, acute exacerbations are further categorized as “triggered” or “idiopathic”, depending on whether an underlying trigger for acute exacerbation is found [13]. As for underlying triggers of acute exacerbation, pharmacotherapy, surgery, and radiotherapy for lung cancer are representative, as well as bacterial and viral infection. No matter how effective the treatment for lung cancer is, once an acute exacerbation of pre-existing IP occurs, it will be fatal to the patient at a high rate. Therefore, it is of utmost importance to choose a therapy that is unlikely to induce acute exacerbation of pre-existing IP.

In this review article, we summarize the current evidence on pharmacotherapy, surgery and perioperative treatment, and radiation therapy for NSCLC with comorbid IP, and discuss future perspectives.

## 2. Pharmacotherapy for NSCLC with Comorbid IP

### 2.1. Treatment Options for Patients with IP

Pharmacotherapy in NSCLC patients with comorbid IP induces acute exacerbation of pre-existing IP in 5–20% of patients, with a mortality rate of 30–50%. Therefore, it is necessary to identify and select drugs that are less likely to cause acute exacerbations. However, since most prospective clinical trials exclude patients with comorbid IP and there are no randomized controlled trials to provide evidence, it is difficult to propose a standard of care as a “guideline”. With this background, the Japanese Respiratory Society issued a “statement” for the treatment of lung cancer with comorbid IP in 2017 [14]. In this statement, cytotoxic drugs are classified into three categories according to the risk of acute exacerbation of IP or pneumonitis (Table 1): “Relatively safe drugs” include platinum-containing drugs, etoposide, paclitaxel, and vinorelbine, “Drugs with no or insufficient data” include Topotecan (nogitecan), pemetrexed, S-1, and docetaxel, and “Drugs not recommended and contraindicated in the package insert” include irinotecan (unconditionally contraindicated in the presence of IP), amrubicin, and gemcitabine (contraindicated if clinical symptoms are present). On the other hand, for the immune checkpoint inhibitors, such as anti-PD-1 antibodies (nivolumab and pembrolizumab) and anti-PD-L1 antibodies (atezolizumab and durvalumab), as well as molecular targeted drugs for various driver gene mutations, there is little data at the time of publication of the statement and no categorization by risk.

### 2.2. Risk Factors for Acute Exacerbations of IP Due to Pharmacotherapy

There have been several studies on risk factors for the development of acute exacerbations of pre-existing IP caused by cytotoxic drugs. Patients with the usual interstitial pneumonia (UIP) pattern, a typical radiological finding of IPF with honeycomb lungs on computed tomography (CT), have more frequent acute exacerbations induced by cytotoxic drugs and more frequent deaths due to acute exacerbations than those with the non-UIP pattern [15]. Patients with lower baseline forced vital capacity (FVC) are also at higher risk of acute exacerbation induced by cytotoxic drugs. Some reports suggest that low FVC was more associated with the risk of acute exacerbation than imaging findings of a UIP pattern [16]. In a post-marketing surveillance of the antifibrotic drug pirfenidone and a subgroup analysis of a randomized phase III trial of nintedanib (INPULSIS) in patients with IPF, acute exacerbations were clearly less frequent in patients with baseline %FVC > 70% [17,18]. However, it should be noted that many NSCLC patients with IP are smokers, and FVC is often overestimated due to emphysematous changes. It has been reported that the higher the modified GAP index, a scoring system for the severity of IPF, the higher the incidence of acute exacerbation of IPF by cytotoxic drugs and the lower the 1-year survival rate [19]. In most prospective studies of cytotoxic drugs conducted in Japan to date, patients with %FVC ≥ 50% and %Diffusing lung capacity for carbon monoxide (DLco) ≥ 30%, who do not require oxygen supplementation at rest, have been included. Therefore, in actual clinical practice, cytotoxic drugs would be considered for these populations. For patients with low pulmonary function (%FVC < 50%) or those who require constant oxygen supplementation, the treatment with cytotoxic drugs may be considered only if performance status and organ function are preserved and the benefits of pharmacotherapy are expected to be sufficient, such as for small-cell lung cancer.

Since pneumonitis induced by immune checkpoint inhibitors (ICIs) presents a variety of imaging findings, distinguishing between “acute exacerbation of pre-existing IP” and “ICI-induced pneumonitis unrelated to pre-existing IP” is more difficult than for cytotoxic anticancer agents. Therefore, in this review article, new interstitial shadows appearing after ICI administration will be lumped together as “ICI-induced pneumonitis”. Several studies showed that the pre-existing IP increases the incidence of ICI-induced pneumonitis. Based on a post-marketing surveillance of nivolumab in Japan (*N* = 3648), abnormal chest CT findings (including emphysematous change and IP) was the independent risk factor for the ICI-induced pneumonitis [20]. The retrospective study of Japanese patients with NSCLC treated with ICI also showed that the incidence of pneumonitis was higher in patients with IP than those in patients without IP (29% vs. 10%, *p* = 0.027) [21]. Furthermore, even “interstitial lung abnormalities”, which refers to mild or subtle radiologic findings incidentally detected on CT without clinical suspicion of IP, is an independent risk factor for the ICI-induced pneumonitis [22]. However, whether there is a difference in the risk of developing ICI-induced pneumonitis depending on the imaging pattern or specific CT findings of pre-existing IP, lung function, or other biomarkers has not been fully investigated at this time.

### 2.3. First-Line Treatment of NSCLC with Comorbid IP

As for the first-line treatment of advanced NSCLC with comorbid IP, there have been six prospective trials reported so far, all of which were single-arm studies (Table 2). Two single-arm phase II studies of carboplatin plus nanoparticle albumin-bound paclitaxel (nab-paclitaxel) have been reported [23,24]. The incidence of acute exacerbation of pre-existing IP was low (4.3–5.6%), and the efficacy was also favorable, with a response rate of 51–56%, median progression-free survival (PFS) of 5.3–6.2 months, and median OS of 15.4 months in both studies. Both trials included a relatively large number of patients (94 and 36 patients, respectively), and these results suggest that this regimen is the current standard of care for untreated NSCLC patients with comorbid IP. The results of two prospective trials have also been reported for carboplatin plus weekly paclitaxel combination therapy [25,26]. The incidence of acute exacerbation induced by this combination therapy ranged from 5.6% to 12.1%. The response rate was 61–70%, with a median progression-free survival of 5.3–6.3 months and a median survival of 10.6–19.8 months, making it a potential treatment option. Furthermore, two prospective trials of carboplatin plus S-1 were conducted in 21 and 33 patients respectively, with an acute exacerbation incidence of 6.1–9.5%, response rate of 33.0–33.3%, median PFS of 4.0–4.8 months, and median OS of 10.4–12.8 months. With these results, carboplatin plus S-1 therapy is another option for first-line treatment [27,28]. The concomitant use of bevacizumab is unlikely to change the risk of triggering the development of acute exacerbations and may be considered in cases where it can be administered [29]. Nintedanib is a multi-kinase inhibitor of platelet-derived growth factor receptor (PDGFR) alpha-beta, fibroblast growth factor receptor (FGFR) 1–3, and vascular endothelial growth factor receptor (VEGFR). Nintedanib is a widely approved and recognized antifibrotic agent for IPF and progressive fibrosing interstitial lung disease in many countries around the world. Nintedanib is expected to be effective not only in slowing down the decline of FVC but also in preventing acute exacerbations. In the INPULSIS study, a randomized phase III trial of IPF, there were significantly fewer adjudicated confirmed/suspected acute exacerbations in the nintedanib group than in the placebo group (1.9% and 4.7%, *p* = 0.010) [18]. In addition, nintedanib is also approved in Europe as a second or subsequent therapy for advanced NSCLC in combination with docetaxel, and is expected to enhance the antitumor effect as an antiangiogenic agent. In Japan, the world’s first randomized phase III trial of carboplatin plus nab-paclitaxel with or without nintedanib in advanced NSCLC with comorbid IPF (J-SONIC trial) was conducted, and is currently under follow-up [30].

In recent years, monotherapy with anti-PD-1/PD-L1 antibodies (nivolumab, pembrolizumab, atezolizumab) or in combination with platinum doublet has become the standard of care for advanced NSCLC patients with PS 0–1, without IP. However, several studies indicated that pre-existing IP increases the risk of ICI-induced pneumonitis [31], and the package insert of various ICIs states that they should be administered with caution in patients with comorbid IP. Therefore, at this time, ICIs should not be used as first-line therapy for patients with IP.

For NSCLC with driver gene mutations/translocations, such as epidermal growth factor receptor (EGFR), anaplastic lymphoma kinase, the BRAF and ROS1 genes, first-line therapy with tyrosine kinase inhibitors (TKIs) targeting the respective gene mutations/translocations is usually recommended. However, when gefitinib-induced pneumonitis appeared with high frequency in Japan and became a social problem, pre-existing IP was identified as an independent risk factor for gefitinib-induced pneumonitis [32]. Since then, special precautions have been required when administering molecular-targeted drugs to driver oncogene-positive NSCLC with comorbid IP. However, since it has been reported that only 0.4% of lung adenocarcinoma patients with EGFR mutations have pre-existing IP [33], there may be few situations in which we actually wonder whether we should administer TKI or not. There are no data on the incidence of pre-existing IP in patients with driver mutations/translocations other than EGFR, or on which types of IP are associated with a higher risk of acute exacerbation induced by TKI, so further study is needed.

### 2.4. Second-Line or Later Treatment of NSCLC with Comorbid IP

Docetaxel and pemetrexed, both standard second-line treatments for NSCLC, have a relatively high incidence of acute exacerbation of pre-existing IP [34,35]. In Japan, a nationwide survey was conducted in 2012 on the details of second-line therapy for lung cancer patients with comorbid IP and the frequency of acute exacerbations for each drug [36]. In this study, the incidence of acute exacerbations of pre-existing IP was 15.3% for docetaxel monotherapy and 28.6% for pemetrexed monotherapy. Therefore, both drugs are not recommended to be administered safely. In the statement for the treatment of lung cancer with comorbid IP published by the Japanese Respiratory Society in 2017, docetaxel and pemetrexed were categorized as “Drugs with no or insufficient data” (Table 1) [14].

S-1 monotherapy is thought to be relatively less likely to induce acute exacerbation of IP based on post-marketing surveillance and retrospective studies, although prospective clinical trials do not exist, and data are scarce. According to the aforementioned nationwide study of second-line pharmacotherapy for lung cancer patients with comorbid IP, the incidence of acute exacerbation of pre-existing IP induced by single-agent S-1 was 0%, although the number of patients studied was small [36].

Nivolumab, an anti-PD-1 antibody, did not induce pneumonitis in a pilot study of six patients with previously treated NSCLC, with mild IP without honeycomb lung and %FVC > 80% [37]. Subsequently, in a phase II study of 18 previously treated NSCLC patients with mild IP selected by the same criteria from four centers, nivolumab induced pneumonitis in 11%. However, all reported ICI-induced pneumonitis were grade 2 and were promptly improved by corticosteroid therapy [38] (Table 1). In addition, these two trials showed remarkable efficacy of nivolumab in NSCLC with comorbid IP, with response rates of 39–50% and disease control rates of 72–100% [36,37]. In a Japanese multicenter, retrospective study comparing the efficacy of single-agent anti-PD-1 antibodies in patients with and without IP, the response rate and disease control rate were both better in the group with IP [39]. As mentioned above, IP (especially IPF) is closely related to microsatellite instability and smoking in its pathogenesis [6]. Therefore, NSCLC with comorbid IP has a higher number of somatic mutations (higher tumor mutation burden) than those without IP, and may be an effective population for ICIs.

On the other hand, one prospective clinical study of ICI in previously treated NSCLC with IP was discontinued due to a high incidence of ICI-induced pneumonitis. TORG1936/AMBITIOUS, a phase II trial of atezolizumab for NSCLC patients with idiopathic, chronic fibrotic IP, with %FVC > 70%, with or without honeycomb lung on chest CT, began enrollment in September 2019. However, due to the high incidence of grade ≥ 3 pneumonitis, this trial was terminated early after 17 patients had been enrolled [40]. In this study, the incidence of ICI-induced pneumonitis was 29% for all grades, 24% for grade 3 or higher, and 6% for grade 5. Of the patients enrolled in the AMBITIOUS study, 35.3% had a UIP pattern and also had a honeycomb lung in 41.2% on the chest CT. The results of the logistic regression analysis suggested that the presence of honeycomb lung may be associated with the development of ICI-induced pneumonitis, although it is not statistically significant. In fact, 57.1% of patients with honeycomb lung in the background developed grade ≥3 pneumonitis, while only 10.0% of patients without honeycomb lung developed grade 1 pneumonitis. Considering the difference in the incidence of pneumonitis between the two nivolumab trials mentioned above and the AMBITIOUS study, as well as the difference in patient selection criteria, it was suggested that the presence of honeycomb lung may be a risk factor for the development of pneumonitis due to ICI. However, since the evaluation of baseline CT findings in the AMBITIOUS study was a post hoc analysis and the number of patients is small, further studies with a larger number of patients are warranted. On the other hand, the median %FVC and %DLco were 85.4% and 54.4% respectively, indicating that the lung function of the patients enrolled in the AMBITIOUS study was relatively preserved. Since many NSCLC patients with comorbid IP are smokers, FVC may not accurately reflect “true” lung function due to emphysematous changes. Therefore, it may be difficult to determine whether ICI can be administered based on the results of pulmonary function tests.

Based on these results, single-agent S-1 is often administered as the standard of care for second-line treatment of NSCLC with comorbid IP in Japanese clinical practice. However, the retrospective studies of cytotoxic drugs as second-line therapy in NSCLC patients with comorbid IP have all shown a 1-year survival rate of at most 10% [34,35]. The efficacy of cytotoxic drugs is limited, and long-term survival can hardly be expected. Thus, for NSCLC patients with comorbid IP, who have a poor prognosis and few treatment options, ICI remains the only existing treatment that offers long-term survival and holds great promise. For appropriate patient selection, large observational and retrospective studies are needed to identify risk factors for ICI-induced pneumonitis by collecting data such as high-resolution CT, pulmonary function tests, and serum biomarkers.

**Table 2 cancers-13-03979-t002:** Key prospective studies in NSCLC with comorbid IP.

Line	Study Design	Phase	Treatment Regimen	N	Incidence of Pneumonitis/Acute Exacerbation of IP	Reference
first line	single arm	2	CBDCA + nab-PTX	94	4.3%	[23]
first line	single arm	2	CBDCA + nab-PTX	36	5.6%	[24]
first line	single arm	Pilot	CBDCA + weekly PTX	18	5.6%	[25]
first line	single arm	2	CBDCA + weekly PTX	35	12.1%	[26]
first line	single arm	Pilot	CBDCA + S-1	21	9.5%	[27]
first line	single arm	2	CBDCA + S-1	33	6.1%	[28]
first line	randomized control trial	3	CBDCA + nab-PTX	120	in progress	[30]
			CBDCA + nab-PTX + Nintedanib	120	in progress	
second line	single arm	Pilot	Nivolumab	6	0.0%	[37]
second line	single arm	2	Nivolumab	18	11.1%	[38]
second line	single arm	2	Atezolizumab	17 (stopped)	29.4%	[40]

Abbreviations: NSCLC, non-small-cell lung cancer; IP, interstitial pneumonia; CBDCA, carboplatin; nab-PTX, nanoparticle albumin-bound paclitaxel; PTX, paclitaxel.

### 2.5. Importance of Follow-Up and Patient Education

Patient selection and drug choice may reduce the risk of acute exacerbations of IP or pneumonitis, but still cannot prevent them completely. Therefore, even if an acute exacerbation or pneumonitis does develop, we need to be able to detect and manage it as early as possible. In actual clinical practice, the initiation of a new pharmacotherapy should be carried out cautiously, with at least the first course hospitalized for around two weeks. Even if started in an outpatient setting, outpatient follow-up should be performed at least once within 1–2 weeks after administration. Carboplatin plus nab-paclitaxel and carboplatin plus weekly paclitaxel, which are used in the first-line treatment of NSCLC with comorbid IP, are regimens that are easy to administer safely because they naturally require weekly visits. In addition, patients should be educated to seek medical attention as early as possible in case of worsening of symptoms such as fever, dyspnea, and cough. Patients should be followed-up approximately every month after treatment.

## 3. Surgery and Perioperative Treatment for NSCLC with Comorbid IP

### 3.1. Surgery

The prognosis of lung cancer with comorbid IP is poor, and indications for pharmacotherapy and radiotherapy are limited. Therefore, surgical resection should be the first option to be considered for patients in operable stages. However, there are two points that should be considered when performing surgical resection for NSCLC with IP. Firstly, the surgery itself may induce an acute exacerbation of IP. According to a study led by the Japanese Association for Chest Surgery, 9.3% of NSCLC patients with IP had acute exacerbations after surgery, and 43.9% of them died [41]. This study also identified seven independent risk factors for acute exacerbations. For example, the risk of developing acute exacerbation of IP varies depending on the surgical procedure, being about four times higher for lobectomy/segmentectomy and about six times higher for bi-lobectomy/total pneumonectomy than for partial pneumonectomy. Based on these risk factors, the following risk scores were also proposed [42]: (1) history of acute exacerbation: 5 points, (2) surgical procedures: 4 points, (3) UIP pattern on chest CT: 4 points, (4) male: 3 points, (5) preoperative steroid use: 3 points, (6) KL-6 ≥ 1000 U/mL: 2 points, and (7) %VC ≤ 80%: 1 point, to calculate the total score. The predicted incidence of acute exacerbation is <10% in the low-risk group (0–10 points), 10–25% in the intermediate-risk group (11–14 points), and >25% in the high-risk group (≥15 points). Secondly, patients with IPF have a poorer long-term prognosis after surgery for NSCLC compared with patients with other IIPs. In a report from Japan, the 5-year survival rate was significantly lower in patients with IPF than in patients with other IIPs (53.2% vs. 22.1%; *p* = 0.0093) [43]. Therefore, whether or not to perform surgery on NSCLC patients with IIPs should be determined by considering the subtype of IIPs and the poor prognosis associated with IPF.

### 3.2. Perioperative Treatment

Postoperative adjuvant chemotherapy is the standard of care for patients with NSCLC without IP, especially in stage N2 and IIIA. However, in patients with comorbid IP, neither postoperative chemotherapy nor radiation is usually indicated because of the risk of acute exacerbation of pre-existing IP.

There are no randomized trials reported so far on perioperative treatment to prevent acute exacerbation of IP after surgery for lung cancer. Although there are reports on the efficacy and safety of civelestat, corticosteroids, macrolides, and urinastine as perioperative treatment, they are all single-arm and with small numbers of case studies, making it difficult to draw definitive conclusions [41,44,45]. The PEOPLE study conducted by the West Japan Oncology Group is a multicenter Phase II study of pirfenidone as a perioperative treatment of NSCLC with comorbid IPF [46]. In this study, acute exacerbations occurred in only 2.8% of 36 patients undergoing surgery for lung cancer with IPF who received pirfenidone in the perioperative period. Based on these results, a randomized phase III study, the PIII-PEOPLE study (NEJ034), is currently underway (UMIN000029411). The patients will be randomized to either the pirfenidone group or the control group for segmentectomy or lobectomy, with subsequent IPF acute exacerbation rate within 30 days after surgery as the primary endpoint. The results of this study are expectedly awaited.

## 4. Radiation Therapy for NSCLC with Comorbid IP

### 4.1. Risk Factors for Acute Exacerbations of IP Due to Radiation Therapy

Thoracic radiotherapy in patients with lung cancer with comorbid IP induces the acute exacerbation of pre-existing IP (or radiation pneumonitis) in 20–30% of patients. Consequently, the risk of death during thoracic radiotherapy is 165.7 times higher in lung cancer patients with IP than in those without IP [47]. Risk factors for acute exacerbations of IP due to radiation therapy include high levels of KL-6, the proportion of normal lungs irradiated ≥5–25 Gy (V5–25), and the mean lung dose (MLD) of normal lungs [48,49]. In addition, a nationwide survey of radiotherapy for lung cancer with IP by the Japanese Radiation Oncology Study Group (JROSG) showed that grading of radiation pneumonitis, FEV1.0, and V30 of the lung were identified as risk factors for acute exacerbation [50]. In a systemic review of stereotactic body radiation therapy (SBRT) for early-stage lung cancer with comorbid IP, V20 ≤ 6.5% and MLD ≤ 4.5 Gy were associated with reduced mortality [51].

### 4.2. Radiation Therapy in Practice

If thoracic radiotherapy in NSCLC patients with comorbid IP induces the acute exacerbation of pre-existing IP, the mortality rate is extremely high [47,52]. Even if IP in the background is mild or subtle and there are no shadows in the irradiated area, radiation therapy can cause fatal acute exacerbation of pre-existing IP. Therefore, if the attending physician judges that the patient has IP based on reasons such as the presence of interstitial shadows on CT and fine crackles on auscultation, it would be safe not to perform thoracic radiotherapy in which the lung field is included in the irradiation field. Radiation therapy for brain metastases or bone metastases in the spine or ribs, where the irradiation field does not include the lung field, may be considered for use if necessary.

On the other hand, in a nationwide survey of radiotherapy for lung cancer with IP by JROSG, only 17% of the respondents answered that radiotherapy for patients with comorbid IP was “unacceptable”, while the remaining 83% answered “acceptable” or “could be a choice” [50]. Thus, there are often disagreements between physician/medical oncologists and radiation oncologists. In the Japanese Respiratory Society statement for the treatment of lung cancer with comorbid IP, radiotherapy itself is not contraindicated [14]. When there is no other alternative, or in the case of an oncologic emergency, palliative radiation therapy, including lung fields, may be a viable option after careful discussion between the patient, attending physician, and radiation oncologist to consider the risks and benefits.

## 5. Conclusions and Future Perspectives

Most of the previous reports on NSCLC with comorbid IP have been in the Japanese population. One possible reason is that respiratory physicians in Japan often diagnose and treat both NSCLC and IP. This may also be due to the fact that the incidence of EGFR-TKI-induced interstitial lung disease and acute exacerbation of IPF tends to be higher in Japanese and East Asians, and there is more interest in these topics. However, as for ICI, the incidence of pneumonitis has been reported to be as high in the Caucasian population as in Japanese or Asian patients [53]. Recently, case series of ICI administered for NSCLC with IP have been published from France [54,55,56], and worldwide interest is gradually increasing, especially in Europe.

The current situation, in which the coronavirus infection 2019 (COVID-19) is still raging around the world, creates new challenges in differentiating acute exacerbations of IP- and drug-induced pneumonitis from COVID-19-related pneumonia. Accurate and rigorous evaluation of pre-existing IP will lead to more appropriate provision of safe and effective treatment to patients with NSCLC. We hope that the topic of “NSCLC with comorbid IP” will attract interest not only from Japan but also from all over the world, and that more data will be accumulated.

## Figures and Tables

**Table 1 cancers-13-03979-t001:** Categorization of each drug in the Japanese Respiratory Society statement for the treatment of lung cancer with comorbid IP.

**Relatively safe drugs**			
platinum-containing drug	etoposide	paclitaxel	vinorelbine
**Drugs with no or insufficient data**			
topotecan	pemetrexed	S-1	docetaxel
**Drugs not recommended and contraindicated in the package insert**			
irinotecan (unconditionally contraindicated in the presence of IP)	amrubicin (contraindicated if clinical symptoms are present)	gemcitabine (contraindicated if clinical symptoms are present)	
**No categorization**			
anti-PD-1 antibodies	anti-PD-L1 antibodies	molecular targeted drugs for various driver mutations	

This table was prepared by modifying the Japanese Respiratory Society statement for the treatment of lung cancer with comorbid IP [14]. Abbreviations: IP, interstitial pneumonia; PD-1, Programmed death 1; PD-L1, Programmed death ligand 1.

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
