# Peer review of "Current Treatment Strategies for Non-Small-Cell Lung Cancer with Comorbid Interstitial Pneumonia"

_cancers, 2021, doi:10.3390/cancers13163979_

Round 1

Reviewer 1 Report

Dear authors of the manuscript “Current Treatment Strategies for Non-Small Cell Lung Cancer with Comorbid Interstitial Pneumonia"

Your article has a very high potential for publication.

In my opinion, only several relatively minor issues need to be addressed before possible release:

Major concerns:

None.

Minor concerns:

  1. Line 193–194: I generally agree with the sentence (“However, since the safety of these ICIs has not been fully elucidated in patients with IP,…. “), but there is a study relating the preexisting interstitial lung disease to the incidence of ICI-related pneumonitis (DOI: 1016/j.lungcan.2018.09.015). Can you adjust the text accordingly.
  2. Line 159: use the range in “OS of 15.4 months“ instead of a single number.

Misspellings:

  1. Check lines 5–14 for typos, missing spaces, etc.
  2. Line 259: change “Since“ to “since“.
  3. Line 306: add a space between “1000“ and “U/ml“.
  4. Line 309: change “>15“ to “≥15“.

Best regards

Author Response

<Minor concern 1>

Line 193–194: I generally agree with the sentence (“However, since the safety of these ICIs has not been fully elucidated in patients with IP,…. “), but there is a study relating the preexisting interstitial lung disease to the incidence of ICI-related pneumonitis (DOI: 1016/j.lungcan.2018.09.015). Can you adjust the text accordingly.

(Response)

Thank you very much for your important comments. Following your advice, I have revised the text in the relevant section as follows.

(Changes)

However, several studies indicated that pre-existing IP increases the risk of ICI-induced pneumonitis [31], and the package insert of various ICIs states that they should be administered with caution in patients with comorbid IP.

<Minor concern 2>

Line 159: use the range in “OS of 15.4 months“, instead of a single number.

(Response)

Thanks for pointing this out. The median OS for both tests was 15.4 months. I have added the following text for clarity.

(Changes)

The incidence of acute exacerbation of pre-existing IP was low (4.3-5.6%), and the efficacy was also favorable, with a response rate of 51-56%, median progression free survival (PFS) of 5.3-6.2 months, and median OS of 15.4 months in both studies.

<Misspellings>

Check lines 5–14 for typos, missing spaces, etc.

Line 259: change “Since“ to “since“.

Line 306: add a space between “1000“ and “U/ml“.

Line 309: change “>15“ to “≥15“.

(Response)

Thank you for your comments. We have checked all of them and made additions and corrections.

Reviewer 2 Report

This review provided valuable information on the treatment strategy of non-small cell lung cancer (NSCLC) patients with interstitial pneumonia (IP). Although there is very limited published literature, the authors have done a great job by describing the effectiveness of pharmacotherapy, surgery & perioperative treatment, and radiation therapy in NSCLC with comorbid IP. The categorization of pharmacotherapy by Japanese respiratory society and clinical trial information of the first and second line therapies are very helpful. The paper will create interest among other scientists and clinicians around the world to generate more data and develop effective strategies to treat NSCLC patients with IP. The review overall is good. However, the authors should come up with one or two graphical representation of the therapies.

Author Response

This review provided valuable information on the treatment strategy of non-small cell lung cancer (NSCLC) patients with interstitial pneumonia (IP). Although there is very limited published literature, the authors have done a great job by describing the effectiveness of pharmacotherapy, surgery & perioperative treatment, and radiation therapy in NSCLC with comorbid IP. The categorization of pharmacotherapy by Japanese respiratory society and clinical trial information of the first and second line therapies are very helpful. The paper will create interest among other scientists and clinicians around the world to generate more data and develop effective strategies to treat NSCLC patients with IP. The review overall is good. However, the authors should come up with one or two graphical representation of the therapies.

(Response)

Thank you very much for your encouraging comments.

As you said, it would be better if we could illustrate a representative case of IP complicated lung cancer. However, advanced NSCLC with IP still has a poor prognosis, even with treatment; long-term survival, as is sometimes the case with NSCLC without IP, is rare, and most patients currently undergoing treatment have an unstable course. Therefore, it is quite difficult to ask for images and other materials for such a manuscript, and we decided not to publish it this time. We would appreciate it if you would take this into consideration and excuse us.